# GRAPH SYNTHETIC OUT-OF-DISTRIBUTION EXPOSURE WITH LARGE LANGUAGE MODELS

## ABSTRACT

Out-of-distribution (OOD) detection in graphs is critical for ensuring model robustness in open-world and safety-sensitive applications. Existing graph OOD detection approaches typically train an in-distribution (ID) classifier on ID data alone, then apply post-hoc scoring to detect OOD instances. While *OOD exposure*—adding auxiliary OOD samples during training—can improve detection, current graph-based methods often assume access to real OOD nodes, which is often impractical or costly. In this paper, we present GOE-LLM, a framework that leverages Large Language Models (LLMs) to achieve OOD exposure on text-attributed graphs without using any real OOD nodes. GOE-LLM introduces two pipelines: (1) identifying pseudo-OOD nodes from the initially unlabeled graph using zero-shot LLM annotations, and (2) generating semantically informative synthetic OOD nodes via LLM-prompted text generation. These pseudo-OOD nodes are then used to regularize ID classifier training and enhance OOD detection awareness. Empirical results on multiple benchmarks show that GOE-LLM substantially outperforms state-of-the-art methods without OOD exposure, achieving up to a 23.5% improvement in AUROC for OOD detection, and attains performance on par with those relying on real OOD nodes for exposure.

## 1 INTRODUCTION

Graph data is widely used to model interactions among entities in social networks, citation networks, transaction networks, recommendation systems, and biological networks Xiao et al. (2020); Zhu et al. (2022); Xu et al. (2021; 2020); Lee et al. (2020). In many practical scenarios, nodes are paired with rich textual attributes—such as user bios, paper abstracts, or product descriptions—leading to *text-attributed graphs* (TAGs) Yang et al. (2021); Yan et al. (2023). These graphs integrate both structural and semantic information, enabling more fine-grained learning and inference tasks. Recently, *out-of-distribution (OOD) detection* on graphs Wu et al. (2023); Ma et al.; Xu et al. (2025b); Zhao et al. (2020); Xu et al. (2025a); Wang et al. (2025); Xu et al. (2025b) has become increasingly studied for safety-critical and open-world applications. The goal is to identify nodes whose distribution significantly deviates from the in-distribution (ID) training classes. This task is particularly relevant in real-world applications where unseen or anomalous entities may appear during inference—such as emerging users in social platforms and new research domains in citation graphs.

**OOD Exposure and Its Applicability to Graphs**. Most existing graph OOD detection methods adopt a semi-supervised, transductive setup in which all nodes are accessible during training, but only a subset of classes is labeled Ma et al.; Song & Wang (2022); Xu et al. (2025a). Training solely on ID data can lead to overconfident predictions on OOD nodes Inkawhich et al. (2021), making subsequent post-hoc OOD scoring less reliable. It can be even worse if OOD instances are structurally or semantically similar to ID data. A widely-used strategy for mitigating this overconfidence is *OOD exposure*, wherein additional OOD samples are incorporated during training Hendrycks et al. (2022; 2018); Zhang et al. (2023); Du et al. (2024). However, existing approaches often rely on real OOD labels—an assumption that is unrealistic in many graph settings Wu et al. (2023), where OOD nodes are elusive or costly to label. In other domains (images, text), recent work has investigated generating *pseudo-OOD* instances Tao et al. (2023); Abbas et al. (2025); Cao et al. (2024); Du et al. (2022); Wang et al. (2020) to reduce reliance on real OOD data. For example, Large Language Models (LLMs) have been used to create OOD proxies for text detection Abbas et al. (2025) or

outlier exposure Cao et al. (2024), yet these techniques are not directly applicable to graph data due to the complexity of node interconnections.

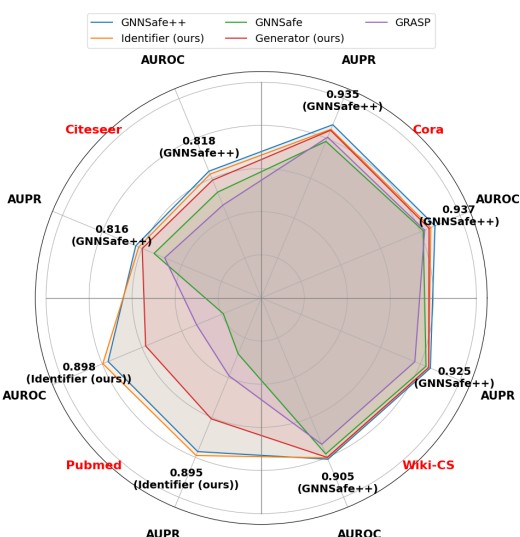

Figure 1: Our graph OOD detection method does not rely on any real OOD nodes for training, yet achieves significantly better OOD detection performance than baseline methods and performs comparably to the approach that uses real OOD nodes (GNNSafe++) for exposure.

**Our Proposal: `GOE-identifier` and `GOE-generator`.** In this paper, we address OOD detection on TAGs by leveraging LLMs to generate OOD supervision signals without using real OOD samples. Specifically, we design two pipelines that inject pseudo-OOD information into training: First: `GOE-identifier`. We randomly sample a small set of unlabeled nodes and prompt an LLM for zero-shot OOD detection. If the LLM concludes that a node does not match any known ID classes, it labels that node as "none," effectively designating it OOD. Despite potential label noise, these *identified* OOD nodes are then used as auxiliary training data to regularize the ID classifier. Second: `GOE-generator`. Instead of annotating existing nodes, we instruct the LLM to *generate* new pseudo-OOD nodes. These are inserted into the original graph to provide OOD signals during regularization training, enhancing the ID classifier's ability to separate OOD from ID classes. Fig. 1 shows that both strategies outperform baselines w/o OOD exposure, while matching approaches with real OOD data.

**`GOE-identifier` vs. `GOE-generator`.** In practice, `GOE-identifier` is convenient when a graph already provides sufficient semantic context, allowing the LLM to reliably mark OOD nodes from unlabeled data. By contrast, `GOE-generator` is preferable if node text is limited or if broad OOD concepts must be introduced. It also applies naturally in inductive scenarios where future nodes—unseen at training time—may be OOD. Meanwhile, `GOE-identifier` targets a transductive setting, since potential OOD nodes must exist beforehand. Overall, `GOE-identifier` provides a lightweight way to detect OOD within the given graph, whereas `GOE-generator` synthesizes new OOD samples when no suitable OOD data or domain knowledge is available in the unlabeled set.

We summarize our key contributions as follow:

- **First Method for LLM-Powered Graph OOD Exposure.** We present a new approach to graph OOD detection with exposure that does not need real OOD labels, leveraging LLMs for pseudo-OOD identification and generation.

- **Exploration of LLM Roles in OOD Exposure**. We propose two approaches for pseudo-OOD supervision: using an LLM as a pseudo-OOD node *identifier* and as a pseudo-OOD node *generator*. `GOE-identifier` is effective in the transductive setting with rich unlabeled node semantics and requires no prior OOD knowledge. `GOE-generator` is applicable to the inductive setting and can introduce novel OOD concepts, but it benefits from a global understanding of OOD semantics.

- **Effectiveness**. Experimental results demonstrate that our method significantly outperforms baselines without OOD exposure and performs similarly to methods that use real OOD nodes for exposure. The code is available at: `https://anonymous.4open.science/r/GOE_LLM-05B7/README.md`.

## 2 RELATED WORK

### 2.1 GRAPH OOD DETECTION

Detecting OOD samples has been extensively investigated in the graph domain. GNNSafe Wu et al. (2023) reveals that standard GNN classifiers inherently exhibit some ability to distinguish OOD nodes, and it proposes an energy-based discriminator trained with a standard classification objective. OODGAT Song & Wang (2022) introduces a feature-propagation mechanism that explicitly

separates inliers from outliers, unifying node classification and outlier detection in a single framework. GRASP Ma et al. demonstrates the benefit of OOD score propagation and theoretical guarantees for post-hoc node-level OOD detection, supplemented by an edge-augmentation strategy. More recently, GNNSafe++ Wu et al. (2023) extends GNNSafe by leveraging real OOD node labels for outlier exposure via an auxiliary regularization objective. However, methods that rely on actual OOD node labels are costly or infeasible, as identifying representative OOD nodes in graphs is non-trivial. Our work addresses the need for OOD exposure *without* real OOD data, specifically on TAGs.

## 2.2 OOD Detection with OOD Exposure

A common strategy to mitigate overconfidence in neural networks is *OOD exposure*, which incorporates OOD examples during training Hendrycks et al. (2018); Yang et al. (2024). In image-based setups, several methods rely on real OOD samples or external datasets, sometimes using mixing strategies to expand the OOD coverage Hendrycks et al. (2022); Zhang et al. (2023). For instance, OECC Papadopoulos et al. (2021) appends a confidence-calibration term to further separate ID and OOD regions, and MixOE Zhang et al. (2023) systematically mixes ID samples with known outliers to smooth the decision boundary. However, such techniques are limited by the availability and quality of genuine OOD data Du et al. (2024), which can be difficult or expensive to acquire in practice—particularly for graph data.

**Pseudo-OOD Generation.** To circumvent the requirement for real OOD examples, recent efforts have explored *pseudo-OOD generation*. VOS Du et al. (2022) synthesizes OOD representations from within the model's latent space, and Vernekar et al. (2019) proposes a method to train an $(n+1)$-class classifier by generating OOD samples in the image domain. Likewise, EOE Cao et al. (2024) uses LLMs to envision outlier concepts for image-based OOD detection, while Abbas et al. (2025) uses LLMs to create high-quality textual proxies for text OOD detection. Despite their successes, these approaches typically overlook graph-specific challenges, where node connectivity and structural information must be modeled alongside semantic content. Indeed, image and text instances are mostly independent, but graph nodes have relational dependencies among neighbors, making it non-trivial to adapt existing pseudo-OOD methods to the graph domain. In this paper, we focus on TAGs and propose using LLMs to identify and generate OOD nodes without relying on any real OOD data, bridging the gap left by prior work that has concentrated on images, text, or real OOD nodes in graphs. A detailed comparison of the above methods is presented in Appendix K.

## 3 Methodology

Here, we introduce GOE-LLM, a framework that integrates pseudo-OOD exposure into TAG learning without requiring real OOD nodes. Fig. 2 summarizes the overall pipeline. Section 3.2 describes the general process of graph OOD exposure, which uses real or pseudo-OOD nodes to regularize the training of the ID classifier. Sections 3.3 and 3.4 detail our proposed methods for graph OOD exposure without real OOD nodes: using an LLM as an OOD node identifier and as an OOD node generator, respectively. Finally, Section 3.5 discusses additional strategies for training a synthetic OOD model to incorporate OOD information into the ID classifier.

## 3.1 Preliminaries

We study node-level OOD detection in TAGs, represented as $G_T = (\mathcal{V}, \mathbf{A}, \mathbf{T}, \mathbf{X})$, where $\mathcal{V} = \{v_1, \ldots, v_n\}$ is the node set with textual attributes $\mathbf{T} = \{t_1, \ldots, t_n\}$. Text embeddings $\mathbf{X} = \{x_1, \ldots, x_n\}$ are derived from $\mathbf{T}$ (e.g., via SentenceBERT Reimers (2019)), and $\mathbf{A} \in \{0, 1\}^{n \times n}$ is the adjacency matrix with $\mathbf{A}[i, j] = 1$ if $v_i$ and $v_j$ are connected.

**Node-level Graph OOD Detection**. We split the nodes into a labeled set $\mathcal{V}_I$ (ID nodes) and an unlabeled set $\mathcal{V}_U$, with $\mathcal{V}_I \cap \mathcal{V}_U = \emptyset$ and $\mathcal{V}_I \cup \mathcal{V}_U = \mathcal{V}$. Each labeled node belongs to one of $K$ known ID classes $Y_I = \{y_1, \ldots, y_K\}$. Unlabeled nodes may be either ID or OOD (i.e., from unknown classes not in $Y_I$). We adopt a semi-supervised, transductive setting: the entire graph is observable at training time, but only $\mathcal{V}_I$ is labeled. The goal is to decide, for each $v_i \in \mathcal{V}_U$, whether it belongs to one of the ID classes or is OOD.

## 3.2 Graph OOD Detection with Pseudo-OOD Exposure

**ID Classifier**. OOD exposure incorporates OOD-like samples during training to tighten the boundary around ID data. We first train a two-layer GCN classifier using only ID-labeled nodes ($\mathcal{V}_I$):

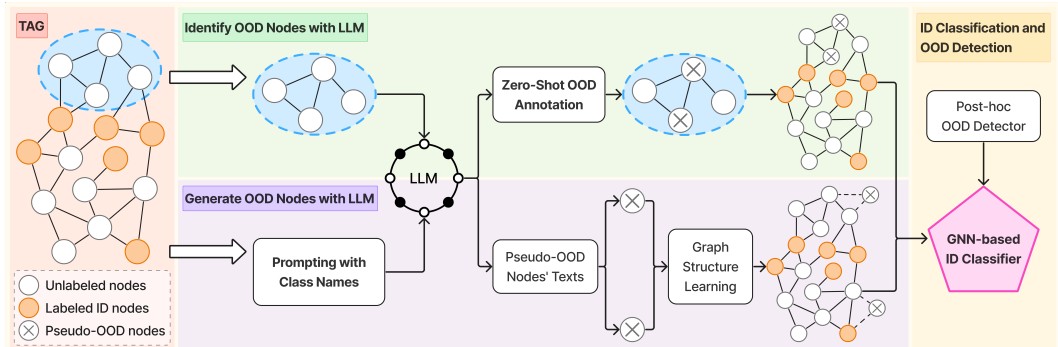

Figure 2: An overview of our framework GOE-LLM. We design two methods for graph OOD exposure without using real OOD nodes: LLM as OOD nodes' identifier and LLM as OOD nodes' generator. The first method employs an LLM to identify potential OOD nodes from the initially fully unlabeled graph. These nodes are then used as pseudo-OOD samples to regularize the training of the ID classifier. The second method prompts the LLM to generate textual descriptions of synthetic OOD nodes, which are subsequently embedded and inserted into the original graph to enrich OOD information during ID classifier training.

$$\mathcal{L}_{\text{sup}} \;=\; -\frac{1}{|\mathcal{V}_I|} \sum_{i \in \mathcal{V}_I} \sum_{k=1}^{K} y_{ik} \, \log \hat{y}_{ik}, \tag{1}$$

where $\hat{y}_{ik}$ is the predicted probability that node $v_i$ belongs to class $k$. After training, any post-hoc OOD scoring function can be applied. One common choice is the negative energy score Liu et al. (2020), where the OOD score for a node $v_i$ with logits $z_i \in \mathbb{R}^K$ is

$$S_{\text{OOD}}(v_i) \;=\; -E(v_i) \;=\; \log \sum_{k=1}^{K} \exp(z_{ik}). \tag{2}$$

A larger $S_{\text{OOD}}(v_i)$ indicates that $v_i$ is more likely OOD.

**Pseudo-OOD Exposure.** To refine the boundary between ID and OOD, we augment the training set with a set $\mathcal{V}_O$ of OOD-like nodes. We then introduce a regularization term $\mathcal{L}_{\text{expo}}$ that enforces contrasting OOD scores for ID nodes and OOD nodes. Specifically, we penalize ID nodes whose OOD scores exceed a margin $s_{\text{id}}$ and OOD nodes whose OOD scores fall below a margin $s_{\text{ood}}$. The overall loss is $\mathcal{L}_{\text{sup}} + \lambda \, \mathcal{L}_{\text{expo}}$, where $\lambda$ balances ID classification accuracy and OOD discriminability.

$$\mathcal{L}_{\text{expo}} = \frac{1}{|\mathcal{V}_I|} \sum_{v_i \in \mathcal{V}_I} \left(\text{ReLU}\left(S_{\text{OOD}}(v_i) - s_{\text{id}}\right)\right)^2 + \frac{1}{|\mathcal{V}_O|} \sum_{v_j \in \mathcal{V}_O} \left(\text{ReLU}\left(s_{\text{ood}} - S_{\text{OOD}}(v_j)\right)\right)^2 \tag{3}$$

where $s_{\text{id}}$ and $s_{\text{ood}}$ are margin parameters, and $\mathcal{V}_O$ can consist of either human-annotated OOD nodes or synthetic ones derived from LLMs.

## 3.3 IDENTIFY OOD NODES WITH LLMS

In this setting, rather than relying on real OOD nodes to train the ID classifier, we assume a more challenging and realistic scenario where there is no access to any OOD nodes or even the names of OOD classes. To address this, we leverage the transductive nature of graph learning and propose a novel approach that instructs LLMs to identify potential OOD nodes directly from the original graph. The identified pseudo-OOD nodes are then used to regularize the training of the ID classifier. Specifically, we first randomly sample a small set of nodes $\mathcal{V}_U^{\text{sampled}}$ from $\mathcal{V}_U$ and let the LLM annotate them. The LLM is only provided with ID knowledge (ID class names) and prompted to determine whether the unlabeled query node belongs to an ID class, using its textual information. We instruct the LLM to output "none" if it determines that the node does not belong to any predefined ID class.

After that, we select the nodes identified by the LLM as OOD to form the pseudo-OOD node set $\mathcal{V}_O$, as shown in Eqn. 4. Using this annotated set $\mathcal{V}_O$, we then train the ID classifier with Eqn. 3.

$$\mathcal{V}_O = \{v \in \mathcal{V}_U^{\text{sampled}} \mid \text{LLM}(\text{prompt}(v, \mathcal{C}_{\text{id}})) = \text{"none"}\} \tag{4}$$

where $\mathcal{C}_{\text{id}}$ denotes the set of ID category names.

### 3.4 GENERATE OOD NODES WITH LLMS

Instead of using LLMs to identify potential OOD nodes in the graph, we propose an alternative approach that leverages LLMs to generate pseudo-OOD nodes and insert them into the original graph for OOD supervision. These generated nodes constitute the OOD instances $\mathcal{V}_O$ as defined in Eqn. 3. In this setting, we assume access only to the label names of all classes, without any real OOD nodes. For each OOD class, we use the LLM to generate $M$ samples, leveraging its inherent large-scale knowledge of the respective class domains. The resulting text descriptions of the pseudo-OOD nodes are denoted by $\mathbf{T}^{\text{ood}}$, and the generation process is formalized in Eqn. 5.

$$\mathcal{V}_O = \{v_m^{\text{ood}} \mid \mathbf{T}_m^{\text{ood}} = \text{LLM}(\text{prompt}(c)), \ 1 \leq m \leq M, \ c \in \mathcal{C}_{\text{ood}}\} \tag{5}$$

where $\mathcal{C}_{\text{ood}}$ denotes the set of OOD category names. By applying SentenceBERT Reimers (2019) to $\mathbf{T}^{\text{ood}}$, we obtain the embeddings of the pseudo-OOD nodes as $\mathbf{X}^{\text{ood}}$. The complete set of node embeddings is then given by $\mathbf{X}_{\text{aug}} = \mathbf{X} \parallel \mathbf{X}^{\text{ood}}$. Optionally, with graph structure learning, we can incorporate the pseudo-OOD nodes into the original graph to better propagate information through the new graph structure, denoted as $\mathbf{A}_{\text{aug}}$. One way to construct $\mathbf{A}_{\text{aug}}$ is by creating edges based on the similarity of node embeddings. Alternatively, a negative sampling-based link prediction task can be performed to infer potential links. Using $\mathbf{X}_{\text{aug}}$ and $\mathbf{A}_{\text{aug}}$, we then train the ID classifier on the augmented graph. This approach reframes OOD detection as an active and generative strategy—rather than a passive inference task—by leveraging textual priors to construct meaningful semantic contrast without requiring real OOD data.

### 3.5 SYNTHETIC OOD MODEL

Thus far, we have proposed using pseudo-OOD nodes to regularize the training of a $K$-class ID classifier and applying post-hoc OOD detectors on top of the well-trained ID classifier for OOD detection. The main advantage of this approach is that it does not require modifying the network architecture of the ID classifier. However, it introduces a trade-off weight $\lambda$ in the loss function. Intuitively, if $\lambda$ is too large, it may degrade ID classification performance. Conversely, if $\lambda$ is too small, the OOD information may not be sufficiently exposed to the ID classifier to improve its OOD awareness. To address this, we provide two alternative approaches that leverage both labeled ID nodes and synthetic noisy OOD nodes to train a model with enhanced OOD awareness.

The first approach involves adding a binary classification layer on top of the standard ID classifier trained using Eqn. 1 to predict OOD scores. Specifically, we first train the ID classifier using labeled ID nodes. Once trained, we freeze the ID classifier and fit the weights of the binary OOD detector using a small set of labeled ID nodes along with the pseudo-OOD samples. The key advantage of this method is that it preserves the ID predictions of the pre-trained classifier while equipping the model with the capability to detect OOD nodes through the additional binary layer. The second approach extends the ID classifier into a $(K + 1)$-way classification model, where the first $K$ classes correspond to the ID classes and the $(K + 1)$-th class represents the OOD class. The model is trained using both labeled ID nodes and pseudo-OOD nodes under a unified cross-entropy loss. The primary advantage of this joint classification approach lies in its flexibility to simultaneously learn accurate ID predictions while distinguishing between ID and OOD nodes, thereby enhancing overall performance.

## 4 EXPERIMENTS

### 4.1 EXPERIMENTAL SETUP

**Datasets** We utilize the following TAG datasets, which are commonly used for node classification: Cora McCallum et al. (2000), Citeseer Giles et al. (1998), Pubmed Sen et al. (2008) and Wiki-CS Mernyei & Cangea (2020). The dataset descriptions are in Appendix B. We follow previous work Song & Wang (2022) by splitting the node classes into ID and OOD classes, ensuring that the number of ID classes is at least two to support the ID classification task. The specific class splits and ID ratios are detailed in Appendix A.

**Training and Evaluation Splits** For each dataset with $K$ ID classes, we use $20 \times K$ of ID nodes for training. Additionally, we randomly select $10 \times K$ of ID nodes along with an equal number of OOD nodes for validation. The test set consists of 500 randomly selected ID nodes and 500 OOD nodes. All experiments are repeated with five random seeds, and results are averaged.

**Baselines** We compare GOE-LLM with baselines: (1) **MSP** Hendrycks & Gimpel (2016), (2) **Entropy**, (3) **Energy** Liu et al. (2020), (4) **GNNSafe** Wu et al. (2023), and (5) **GRASP** Ma et al., all of which are post-hoc OOD detection methods without exposure. We also include **OE** Hendrycks et al. (2018) and **GNNSafe++** Wu et al. (2023), which leverage real OOD samples for exposure.

**LLM-Powered OOD Exposure.** We use GPT-4o-mini for pseudo-OOD identification and generation. For node identification, we randomly sample 200 unlabeled nodes per dataset and prompt the LLM to classify them as either ID or OOD. Nodes predicted as OOD are used as pseudo-OOD exposure data. For pseudo-OOD generation, we prompt the LLM to generate $10 \times K_{\text{ood}}$ nodes, where $K_{\text{ood}}$ denotes the number of OOD classes. The generated node texts are embedded using SentenceBERT Reimers (2019), and the resulting embeddings are concatenated with the original graph embeddings. We leave the exploration of more advanced graph structure learning methods for constructing the augmented graph to future work. Details on the efficiency of our method are provided in Appendix F.

**Implementation details** For fair comparison, all ID classifiers are implemented using 2-layer GCNs with hidden dimension 32. We use Adam optimizer with learning rate 0.01, dropout rate 0.5, and weight decay of 5e-4. For all OOD exposure methods, the trade-off weight $\lambda$ in the loss function is selected from $\{0.01, 0.05\}$ based on the results of the validation set. For all methods, we set the maximum number of training epochs to 200 and apply early stopping if the sum of AUROC and ID ACC does not improve for 20 epochs. All experiments are conducted on hardware equipped with an NVIDIA GeForce RTX 4080 SUPER GPU.

**Evaluation Metrics** For the ID classification task, we use classification accuracy (ID ACC) as the evaluation metric. For the OOD detection task, we employ three standard metrics Wu et al. (2023): the area under the ROC curve (AUROC), the precision-recall curve (AUPR), and the false positive rate when the true positive rate reaches 95% (FPR@95). In all experiments, OOD nodes are considered positive cases. Detailed descriptions of these metrics are provided in Appendix C.

### 4.2 MAIN RESULTS

Table 1 presents the performance of various OOD detection methods across four datasets. From the results, we make several key observations:

**Effectiveness of LLM-driven OOD Exposure.** Both variants of GOE-LLM —GOE-identifier and GOE-generator —consistently outperform all methods that do not utilize OOD exposure. For example, on the Pubmed dataset, GOE-identifier achieves an AUROC of 0.8985, significantly surpassing GRASP (0.6627), the best-performing method without OOD exposure, resulting in a relative improvement of over 23.5%. This demonstrates that the pseudo-OOD nodes identified by the LLM provide meaningful supervision for learning precise decision boundaries in open-world settings. However, the improvement on the Wiki-CS dataset is much smaller, suggesting that the effectiveness of pseudo-OOD exposure depends on the dataset's inherent difficulty and the extent of distributional shift. Due to space constraints, the standard deviation results are provided in Appendix D.

**Comparable Performance to Real OOD Supervision.** Remarkably, GOE-LLM achieves performance comparable to OE and GNNSafe++, both of which use real OOD nodes annotated by humans. On the Pubmed dataset, GOE-LLM even surpasses OE and GNNSafe++ in terms of AUROC and AUPR, despite relying solely on noisy pseudo-OOD nodes. This demonstrates that LLMs can serve as a practical alternative to costly OOD data curation. We attribute this strong performance to the LLM's ability to synthesize semantically coherent yet distributionally distinct samples that effectively emulate the characteristics of true OOD data.

**ID Classification is Maintained.** A common concern with OOD exposure is its potential to degrade ID classification performance due to over-regularization. However, our results show that GOE-LLM maintains strong ID classification accuracy across all datasets. For example, on Citeseer, GOE-generator reaches 0.8552 ID accuracy, outperforming all other baselines. This indicates that pseudo-OOD nodes do not dilute the model's ability to learn discriminative features for the ID task.

**Insights on LLM Annotations.** Although the OOD nodes identified by LLMs are relatively noisy (as shown in Section 4.4), using these noisy nodes to regularize the training of the ID classifier still yields

Table 1: Performance comparison (best in **bold**, second-best in underline) of different methods on ID classification and OOD detection tasks. GOE-LLM achieves comparable performance to methods using real OOD nodes, while requiring no real OOD data and significantly outperforming methods without OOD exposure.

| Model | Methods | Cora | | | | Citeseer | | | | Pubmed | | | | Wiki-CS | | | |
|---|---|---|---|---|---|---|---|---|---|---|---|---|---|---|---|---|---|
| | | ID ACC ↑ | AUROC ↑ | AUPR ↑ | FPR95 ↓ | ID ACC ↑ | AUROC ↑ | AUPR ↑ | FPR95 ↓ | ID ACC ↑ | AUROC ↑ | AUPR ↑ | FPR95 ↓ | ID ACC ↑ | AUROC ↑ | AUPR ↑ | FPR95 ↓ |
| No OOD Exposure | MSP | 0.8748 | 0.8414 | 0.8506 | 0.6428 | 0.8480 | 0.7466 | 0.7500 | 0.7976 | 0.8776 | 0.6591 | 0.6623 | 0.8908 | 0.8648 | 0.7772 | 0.7851 | 0.7440 |
| | Entropy | 0.8800 | 0.8471 | 0.8549 | 0.6300 | 0.8480 | 0.7655 | 0.7603 | 0.7244 | 0.8776 | 0.6591 | 0.6623 | 0.8908 | 0.8640 | 0.7823 | 0.7891 | 0.7440 |
| | Energy | 0.8788 | 0.8580 | 0.8648 | 0.5928 | 0.8504 | 0.7757 | 0.7754 | 0.7256 | 0.8876 | 0.5861 | 0.5919 | 0.9296 | 0.8648 | 0.7983 | 0.8056 | 0.7320 |
| | GNNSafe | 0.8800 | 0.9073 | 0.9073 | 0.4084 | 0.8504 | 0.7654 | 0.7697 | 0.8004 | **0.8916** | 0.5954 | 0.6403 | 0.8928 | 0.8752 | 0.8915 | 0.9144 | 0.7928 |
| | GRASP | 0.8800 | 0.9111 | 0.9034 | 0.3820 | 0.8460 | 0.7329 | 0.7429 | 0.8560 | 0.8908 | 0.6627 | 0.6956 | 0.8780 | 0.8768 | 0.8674 | 0.8864 | 0.7860 |
| Ours | GOE-identifier | 0.8780 | 0.9264 | 0.9233 | 0.3268 | 0.8444 | 0.8107 | 0.8082 | 0.7104 | 0.8636 | **0.8985** | **0.8955** | **0.5108** | 0.8764 | 0.9014 | 0.9212 | 0.7728 |
| | GOE-generator | 0.8792 | 0.9221 | 0.9208 | 0.3372 | **0.8552** | 0.7956 | 0.7994 | 0.7328 | 0.8820 | 0.7908 | 0.8035 | 0.7780 | 0.8752 | 0.9002 | 0.9214 | 0.7288 |
| Real OOD Exposure | OE | **0.8820** | 0.9362 | 0.9392 | 0.3196 | 0.8384 | **0.8348** | **0.8254** | **0.6100** | 0.8512 | 0.8236 | 0.8314 | 0.7236 | **0.8832** | **0.9166** | 0.9171 | **0.4248** |
| | GNNSafe++ | 0.8792 | **0.9371** | **0.9347** | **0.2944** | 0.8464 | 0.8180 | 0.8157 | 0.6832 | 0.8760 | 0.8852 | 0.8857 | 0.5568 | 0.8816 | 0.9048 | **0.9254** | 0.7668 |

results on par with those obtained using real OOD nodes. The key intuition is that, while LLM-based annotations may be noisy, they are not arbitrary. In fact, they reflect a distributionally-aware semantic prior: nodes misclassified as OOD by the LLM often lie near the ID-OOD boundary and can serve as hard negatives. This is evident in our results, where even imprecise OOD exposure improves downstream detection performance significantly. This supports the hypothesis that OOD exposure does not need to be perfect to be effective—it merely needs to be informative enough to delineate boundaries in the feature space.

**Overall Impact.** Taken together, our findings suggest that LLM-driven pseudo-OOD exposure is a promising and scalable direction for graph OOD detection. It enables label-free OOD supervision, maintains strong ID classification, and yields competitive or superior results compared to both non-exposure and real-exposure methods.

## 4.3 IS THE CURRENT GRAPH OOD DETECTION SETTING PRACTICAL?

Since there is no established graph-specific OOD detection benchmark that models real distributional shifts among nodes, most existing work on node-level graph OOD detection simply assumes that nodes from randomly selected classes are designated as OOD. However, this setting has notable limitations. For example, if the ID classes are more heterogeneous or closely resemble the OOD classes, the ID classifier may fail to learn a well-defined decision boundary, resulting in high softmax confidence scores for OOD nodes. Additionally, if the set of ID classes spans a broad and diverse range of features, the model may naturally assign high confidence to OOD nodes, violating the common assumption that OOD nodes should receive lower confidence scores. Therefore, using real or pseudo-OOD nodes can provide a more realistic and effective training signal for OOD detection. By explicitly exposing the model to samples that are semantically or structurally different from the ID distribution, we can help the classifier better distinguish between ID and OOD nodes. This leads to more calibrated confidence estimates and improved robustness under open-world scenarios.

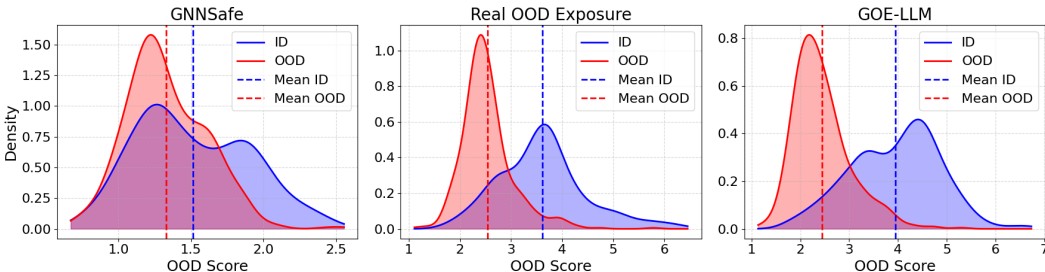

Figure 3: OOD score distributions of GNNSafe, GNNSafe++, and GOE-LLM on the Pubmed dataset. GOE-LLM, despite not using any real OOD nodes, achieves distributional separation comparable to the real OOD exposure method and significantly better separability than GNNSafe.

To demonstrate this, we visualize the OOD scores on the Pubmed dataset for GNNSafe, GOE-LLM, and GNNSafe++. As shown in Fig. 3, compared to the method without OOD exposure, our approach assigns notably lower OOD scores to ID nodes and higher scores to OOD nodes, resulting in a more distinct separation between the two groups.

## 4.4 ZERO-SHOT OOD ANNOTATION

We present the zero-shot OOD detection performance of the LLM. The prompt used for zero-shot OOD node identification is provided in Prompt 4.4. For each dataset, we prompt the LLM to determine whether each node in the test set (comprising 500 ID nodes and 500 OOD nodes) belongs to one of the predefined ID categories; if not, the node is considered an OOD instance. We then obtain binary OOD predictions from the LLM and compute its accuracy. The results are reported in Table 2. We use accuracy as the sole evaluation metric for the LLM's zero-shot OOD annotation performance, since the LLM does not produce soft OOD scores (e.g., as defined in Eqn. 2), but rather outputs hard binary decisions (0 or 1).

---

**PROMPT: Zero-Shot OOD Node Identification**

```
As a research scientist, your task is to analyze and classify {object} based on their
main topics, meanings, background, and methods.

Please first read the content of the {object} carefully.  Then, identify the {object}'s
key focus.  Finally, match the content to one of the given categories:

[Category 1, Category 2, Category 3, ...]

Given the current possible categories, determine if it belongs to one of them.  If so,
specify that category; otherwise, say "none".

[Insert {Object} Content Here]
```

|  | **Cora** | **Citeseer** | **Pubmed** | **Wiki-CS** |
|---|---|---|---|---|
| Zero-shot OOD annotation | 0.7190 | 0.7260 | 0.8410 | 0.7470 |

Table 2: Accuracy of using the LLM to identify whether unlabeled nodes are OOD. Although the LLM-identified pseudo-OOD nodes are noisy, they still enable effective OOD exposure.

## 4.5 SYNTHETIC DATA MODEL

In this section, we combine labeled ID nodes and pseudo-OOD nodes to train the synthetic data models, as described in Section 3.5. Specifically, we randomly select $20 \times K$ ID nodes and pseudo-OOD nodes annotated by the LLM. In this way, we use the same number of ID and OOD nodes to train the synthetic data models.

Table 3: Performance comparison (best highlighted in bold) of different synthetic data models on ID classification and OOD detection tasks.

| Methods | Cora | | | | Citeseer | | | | Pubmed | | | | Wiki-CS | | | |
|---|---|---|---|---|---|---|---|---|---|---|---|---|---|---|---|---|
| | ACC ↑ | AUROC ↑ | AUPR ↑ | FPR95 ↓ | ACC ↑ | AUROC ↑ | AUPR ↑ | FPR95 ↓ | ACC ↑ | AUROC ↑ | AUPR ↑ | FPR95 ↓ | ACC ↑ | AUROC ↑ | AUPR ↑ | FPR95 ↓ |
| GOE-LLM | 0.8780 | 0.9264 | 0.9233 | 0.3268 | 0.8444 | 0.8107 | 0.8082 | 0.7104 | 0.8636 | 0.8985 | 0.8955 | 0.5108 | 0.8764 | 0.9014 | 0.9212 | 0.7728 |
| $(K+1)$-Classifier | 0.8716 | 0.9138 | 0.9185 | 0.4224 | 0.8336 | 0.8189 | 0.8200 | 0.6648 | 0.8840 | 0.9060 | 0.8968 | 0.3980 | 0.8684 | 0.8015 | 0.7923 | 0.6648 |

For the first approach, we can add a binary classification layer on top of the output features of the ID classifier to predict the OOD score $z_{\text{ood}} = \mathbf{w}^\top \phi(x)$, where $\mathbf{w} \in \mathbb{R}^h$, and $x$ is the output of the hidden layer from the GNN-based ID classifier. The OOD score of node $v_i$ is then defined as $S_{\text{OOD}}(v_i) = \sigma(z_{\text{ood}})$, where $\sigma(\cdot)$ denotes the sigmoid function. In the second approach, we train a $(K+1)$-class classifier and define the softmax probability of the $(K+1)$-th class as the OOD score. However, the performance of the first approach is not satisfactory in the current setting; therefore, we only report the results of the $(K+1)$-class classifier. The results are presented in Table 3. From the results, we observe that using Eqn. 3 to regularize the training of the ID classifier does not degrade ID classification performance. At the same time, it significantly improves OOD detection performance compared to the baselines without OOD exposure. Furthermore, the $(K+1)$-class classifier generally achieves performance comparable to that of the regularization method.

## 4.6 HOW MANY SYNTHETIC OOD NODES DO WE NEED?

In this section, we prompt the LLM to generate different numbers of pseudo-OOD nodes. The prompt used for OOD node generation is provided in Appendix E. Table 4 presents the ID classification and OOD detection performance on the Pubmed dataset (which contains 19,717 nodes) using different numbers of generated pseudo-OOD nodes. From the results, we can see that using more pseudo-OOD nodes improves OOD detection performance. When the number of generated OOD nodes

| Number of Pseudo-OOD Nodes | ID ACC ↑ | AUROC ↑ | AUPR ↑ | FPR@95 ↓ |
|---|---|---|---|---|
| 0 | **0.8916** | 0.5954 | 0.6403 | 0.8928 |
| 2 | 0.8864 | 0.6692 | 0.6961 | 0.8708 |
| 3 | 0.8896 | 0.7401 | 0.7566 | 0.8100 |
| 5 | 0.8796 | 0.7674 | 0.7857 | 0.8036 |
| 10 | 0.8820 | 0.7908 | 0.8035 | 0.7780 |
| 20 | 0.8856 | **0.8043** | **0.8073** | **0.7400** |

Table 4: Performance comparison using different numbers of generated pseudo-OOD nodes for OOD exposure on the Pubmed dataset. Even a small number of pseudo-OOD nodes significantly improves OOD detection performance.

reaches around 10, the performance nearly converges to a value that is significantly higher than that achieved without OOD exposure. This suggests that even a small number of pseudo-OOD nodes can provide meaningful OOD exposure during training, helping the model learn a more accurate decision boundary between ID and OOD nodes. Moreover, it shows that LLM-generated pseudo-OOD nodes offer an efficient and lightweight substitute for real OOD data. Another observation is that, among all cases, the ID classification accuracy is highest when no OOD exposure is performed. However, the degradation in ID classification performance is negligible when pseudo-OOD nodes are used to regularize the training of the ID classifier. This further demonstrates that the trade-off described in Section 3.2 is favorable, as pseudo-OOD exposure significantly improves OOD detection performance while having minimal impact on ID classification accuracy.

## 4.7 COMPARISON ACROSS LLM VARIANTS

We experiment with the open-source model DeepSeek-V3 and the older LLM GPT-3.5-Turbo. The results in Table 5 show only mild performance degradation with GPT-3.5 compared to GPT-4o-mini. Moreover, using open-source LLMs for OOD node identification and generation yields comparable detection performance, demonstrating that our approach is robust and reproducible with open-source alternatives. The cost of querying the LLM for OOD annotations is reported in Appendix F.

| Method | Cora | | | | Citeseer | | | |
|---|---|---|---|---|---|---|---|---|
| | ID ACC↑ | AUROC↑ | AUPR↑ | FPR95↓ | ID ACC↑ | AUROC↑ | AUPR↑ | FPR95↓ |
| GOE-identifier (DeepSeek-V3) | 0.8812 | 0.9288 | 0.9256 | 0.3136 | 0.8520 | 0.7902 | 0.7873 | 0.7612 |
| GOE-generator (DeepSeek-V3) | 0.8796 | 0.9244 | 0.9224 | 0.3252 | 0.8476 | 0.7815 | 0.7889 | 0.7696 |
| GOE-identifier (GPT-3.5-Turbo) | 0.8836 | 0.9202 | 0.9187 | 0.3340 | 0.8512 | 0.7978 | 0.7946 | 0.7488 |
| GOE-generator (GPT-3.5-Turbo) | 0.8796 | 0.9185 | 0.9173 | 0.3384 | 0.8512 | 0.7923 | 0.7978 | 0.7648 |

Table 5: Performance comparison of GOE-LLM with different LLMs on Cora and Citeseer datasets.

## 5 CONCLUSION

In this paper, we propose GOE-LLM, a novel framework for OOD detection on TAGs that eliminates the reliance on real OOD data for exposure by leveraging LLMs. By designing two exposure strategies—LLM-based OOD node identification and OOD node generation—we enable label-efficient, scalable, and effective OOD exposure. Extensive experiments across diverse datasets demonstrate that GOE-LLM achieves strong performance compared to methods relying on real OOD data, and significantly surpasses existing methods without OOD exposure. Future work could explore more advanced prompting strategies to improve the quality of pseudo-OOD samples. Additionally, incorporating adaptive graph structure learning tailored to generated nodes may further boost performance. Extending GOE-LLM to other graph learning tasks—such as graph-level OOD detection and OOD detection on dynamic TAGs—also presents a promising direction.

**Broader Impact**. On the positive side, this work provides a practical and generalizable solution to improve model reliability and robustness in open-world scenarios where labeled data is scarce or dynamic changes are frequent. This has the potential to enhance safety and trustworthiness in real-world deployments. On the other hand, our reliance on LLM-generated supervision introduces challenges around semantic validity and bias. The quality and fairness of the pseudo-OOD labels depend on the pretrained LLMs, which may inherit societal or domain-specific biases.

**Limitations**. First, the effectiveness of pseudo-OOD nodes relies on the LLM's zero-shot accuracy; inaccurate outputs may introduce noisy supervision. Second, while the proposed method markedly improves OOD detection performance on many graph datasets, it is applicable only to TAGs. For graphs without textual information, the method may be ineffective, highlighting the need for more generalizable OOD exposure techniques. Further discussion is provided in Appendix H and I.

## ETHICS STATEMENT

We adhere to the ICLR Code of Ethics and take full responsibility for this work. Our experiments use public, de-identified text-attributed graph datasets (Cora, Citeseer, Pubmed, Wiki-CS) under their licenses; no PII or new human-subject data were collected. LLMs were used only for pseudo-OOD identification/generation and language editing, and all LLM-assisted text was reviewed by the authors. We recognize potential misuse and mitigate it by releasing research-oriented code/prompts, documenting limitations, and avoiding person-level datasets. To support fairness and transparency, we report class splits, metrics, seeds, and ablations across datasets. Compute and environmental impact are modest (small GNNs; limited training-time LLM calls; no LLMs at inference). Conflicts of interest, if any, are disclosed per venue policy.

## REPRODUCIBILITY STATEMENT

We provide complete details to facilitate replication. The formal description of our methods (GOE-identifier, GOE-generator) and training procedures appears in Section 3 and Section 4, including datasets, class partitions, and evaluation protocol. Dataset choices (Cora, Citeseer, Pubmed, Wiki-CS) and ID/OOD class splits are described in the Experimental Setup and appendices. We specify training/validation/test splits and report averages over five random seeds. Baselines and comparison settings are enumerated alongside our methods to ensure parity. Prompts used for LLM-based pseudo-OOD identification and generation are included in Section 4.4 and the appendix for exact reuse. Additional ablations (e.g., varying the number of pseudo-OOD nodes) and efficiency analyses are provided in the appendices. Our anonymous code repository is linked in the main paper for end-to-end reproduction (data preprocessing, training scripts, and evaluation).

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

# APPENDIX: GRAPH SYNTHETIC OUT-OF-DISTRIBUTION EXPOSURE WITH LARGE LANGUAGE MODELS

## A  ID AND OOD SPLIT

Table 6: ID classes and ID ratio for different datasets.

| Dataset | ID class | ID ratio |
|---|---|---|
| Cora | [2, 4, 5, 6] | 47.71% |
| Citeseer | [0, 1, 2] | 55.62% |
| WikiCS | [1, 4, 5, 6] | 38.79% |
| Pubmed | [0, 1] | 60.75% |

## B  DATASET DESCRIPTIONS

**Cora**  The Cora dataset McCallum et al. (2000) contains 2,708 scientific publications categorized into seven research topics: case-based reasoning, genetic algorithms, neural networks, probabilistic methods, reinforcement learning, rule learning, and theory. Each node represents a paper, and edges correspond to citation links between papers, forming a graph with 5,429 edges.

**CiteSeer**  The CiteSeer dataset Giles et al. (1998) consists of 3,186 scientific articles classified into six research domains: Agents, Machine Learning, Information Retrieval, Databases, Human-Computer Interaction, and Artificial Intelligence. Each node represents a paper, with node features extracted from the paper's title and abstract. The graph is constructed based on citation relationships among the publications.

**WikiCS**  WikiCS Mernyei & Cangea (2020) is a Wikipedia-based graph dataset constructed for benchmarking graph neural networks. Nodes correspond to articles in computer science, divided into ten subfields serving as classification labels. Edges represent hyperlinks between articles, and node features are derived from the corresponding article texts.

**PubMed**  The PubMed dataset Sen et al. (2008) comprises scientific articles related to diabetes research, divided into three categories: experimental studies on mechanisms and treatments, research on Type 1 Diabetes with an autoimmune focus, and Type 2 Diabetes studies emphasizing insulin resistance and management. The citation graph connects related papers, with node features derived from medical abstracts.

## C  EVALUATION METRICS

We use the following metrics to evaluate in-distribution (ID) classification and out-of-distribution (OOD) detection performance, which are widely used OOD detection research Dong et al. (2024); Li et al. (2025); Qin et al. (2025):

**Accuracy (ACC)**  Measures the proportion of correctly classified ID nodes:

$$\text{ACC} = \frac{1}{|\mathcal{D}_{\text{ID}}|} \sum_{x_i \in \mathcal{D}_{\text{ID}}} \mathbb{I}[\hat{y}_i = y_i], \tag{6}$$

where $\hat{y}_i$ is the predicted class label and $y_i$ is the true class label.

**Area Under the ROC Curve (AUROC)**  Evaluates how well the OOD detector ranks OOD nodes higher than ID nodes based on their OOD scores. It is defined as:

$$\text{AUROC} = \mathbb{P}\left(s_{\text{OOD}}(x_{\text{OOD}}) > s_{\text{OOD}}(x_{\text{ID}})\right), \tag{7}$$

where $s_{\text{OOD}}(\cdot)$ denotes the OOD score function.

**Area Under the Precision-Recall Curve (AUPR)**    Measures the area under the curve defined by precision and recall:

$$\text{Precision} = \frac{\text{TP}}{\text{TP} + \text{FP}}, \tag{8}$$

$$\text{Recall} = \frac{\text{TP}}{\text{TP} + \text{FN}}, \tag{9}$$

$$\text{AUPR} = \int_0^1 \text{Precision}(r)\, dr, \tag{10}$$

where OOD nodes are treated as the positive class, and $r$ denotes recall.

**False Positive Rate at 95% True Positive Rate (FPR@95)**    Indicates the fraction of ID samples incorrectly predicted as OOD when the true positive rate (TPR) on OOD samples is 95%:

$$\text{FPR@95} = \left. \frac{\text{FP}_{\text{ID}}}{\text{FP}_{\text{ID}} + \text{TN}_{\text{ID}}} \right|_{\text{TPR}=0.95}, \tag{11}$$

where FP and TN are false positives and true negatives on ID data, respectively.

# D    STANDARD DEVIATION RESULTS

Table 7: Standard deviation results for various models across ID classification and OOD detection metrics. All values are in percentages.

| Model | Methods | Cora | | | | Citeseer | | | | Pubmed | | | | Wiki-CS | | | |
|---|---|---|---|---|---|---|---|---|---|---|---|---|---|---|---|---|---|
| | | ACC | AUROC | AUPR | FPR95 | ACC | AUROC | AUPR | FPR95 | ACC | AUROC | AUPR | FPR95 | ACC | AUROC | AUPR | FPR95 |
| No OOD Exposure | MSP | 2.46 | 1.90 | 1.47 | 7.68 | 1.39 | 2.85 | 3.04 | 5.38 | 3.06 | 2.12 | 3.02 | 2.37 | 2.73 | 3.65 | 4.19 | 6.74 |
| | Entropy | 2.18 | 2.78 | 1.87 | 8.91 | 1.39 | 2.59 | 2.80 | 6.31 | 3.06 | 2.12 | 3.02 | 2.37 | 2.71 | 3.33 | 3.71 | 7.10 |
| | Energy | 2.17 | 2.18 | 1.58 | 9.95 | 1.61 | 2.54 | 2.27 | 5.68 | 1.13 | 8.70 | 9.23 | 2.82 | 2.67 | 5.70 | 4.92 | 13.74 |
| | GNNSafe | 2.18 | 1.74 | 2.12 | 10.52 | 1.61 | 2.58 | 2.36 | 5.91 | 1.23 | 14.34 | 13.08 | 8.26 | 2.63 | 0.80 | 0.90 | 16.60 |
| | GRASP | 2.18 | 1.88 | 1.34 | 13.29 | 1.62 | 2.88 | 2.82 | 6.92 | 1.32 | 12.34 | 11.76 | 5.70 | 2.92 | 2.58 | 3.18 | 12.96 |
| Ours | GOE-identifier | 2.53 | 0.88 | 1.42 | 2.24 | 1.65 | 2.01 | 1.85 | 7.66 | 2.03 | 2.48 | 3.02 | 7.69 | 2.20 | 0.90 | 1.26 | 11.86 |
| | GOE-generator | 2.18 | 1.67 | 2.12 | 7.03 | 1.49 | 1.31 | 1.58 | 7.12 | 2.69 | 6.86 | 5.62 | 7.79 | 2.66 | 1.22 | 1.09 | 1.62 |
| Real OOD Exposure | OE | 1.86 | 1.19 | 1.03 | 6.39 | 1.60 | 2.57 | 2.24 | 11.08 | 1.88 | 5.17 | 5.27 | 14.78 | 1.40 | 1.64 | 1.71 | 9.08 |
| | GNNSafe++ | 2.57 | 0.93 | 1.46 | 3.84 | 1.47 | 2.08 | 2.22 | 8.69 | 1.26 | 3.83 | 3.45 | 1.13 | 2.41 | 1.04 | 1.32 | 12.27 |

# E    PROMPT FOR OOD NODE GENERATION

**PROMPT: LLM-Based OOD Node Generation**

```
Please generate {num_generated_samples} {object}(s) belonging to the
category '{category_name}', including title and abstract.
```

**Output Format:**

```
• Title:  <Generated Title>
• Abstract:  <Generated Abstract>
```

# F    EFFICIENCY AND COST ANALYSIS.

During training, GOE-identifier involves approximately 200 LLM API calls per dataset for node annotation, each taking ∼1 second per call. GOE-generator typically involves generating around 10 synthetic nodes per OOD class, resulting in approximately 10–50 API calls per dataset, each call taking ∼1 second.

However, our method is highly efficient at inference. Unlike directly using an LLM for OOD detection, it relies solely on the trained GNN-based ID classifier, yielding significantly faster inference. As shown in Table 8, no LLM components are involved, resulting in inference speeds comparable to post-hoc OOD detection methods.

The monetary cost of querying the LLM (GPT-4o-mini) for annotations is reported in Table 9. By default, GOE-identifier uses the LLM to **annotate 200 nodes** per dataset, while GOE-generator **generates** $10 \times K$ **nodes** per dataset. For example, on the Cora dataset, GOE-identifier costs **$0.0236** in total, and GOE-generator costs **$0.007** in total.

|  | **GNNSafe** | **GOE-identifier** | **GOE-generator** |
|---|---|---|---|
| Training time (s) | 0.45 | 0.56 | 0.50 |
| Inference time (s) | 0.0023 | 0.0024 | 0.0024 |

Table 8: Training and inference times of GNNSafe, GOE-identifier, and GOE-generator on the Citeseer dataset.

|  | **GOE-identifier** | **GOE-generator** |
|---|---|---|
| Total cost | $0.02357 / 200 nodes | $0.00702 / 30 nodes |
| Cost per node | $0.000118 / node | $0.000234 / node |

Table 9: Monetary cost of GOE-identifier and GOE-generator when using GPT-4o-mini for annotations on the Cora dataset.

## G  COMPARISON WITH ZERO-SHOT LLM DETECTION

In the following table, we also report the results of directly using binary OOD predictions from the LLM to compute OOD detection metrics (AUROC, AUPR, FPR95). This direct comparison demonstrates that our proposed synthetic OOD exposure strategy significantly improves OOD detection performance over zero-shot LLM detection alone.

|  | **Cora** | | | | **Citeseer** | | | |
|---|---|---|---|---|---|---|---|---|
| **Method** | ID ACC↑ | AUROC↑ | AUPR↑ | FPR95↓ | ID ACC↑ | AUROC↑ | AUPR↑ | FPR95↓ |
| LLM-zero-shot | 0.6032 | 0.7310 | 0.7982 | 1.0000 | 0.4400 | 0.7268 | 0.8087 | 1.0000 |
| GOE-LLM | 0.8780 | 0.9264 | 0.9233 | 0.3268 | 0.8444 | 0.8107 | 0.8082 | 0.7104 |

Table 10: Comparison of zero-shot LLM OOD detection versus GOE-LLM with synthetic OOD exposure on Cora and Citeseer datasets

## H  DISCUSSION ON POTENTIAL OVERLAP BETWEEN LLM TRAINING DATA AND EVALUATION DATASETS.

GPT-4o-mini is a proprietary model, and detailed training corpus information is not publicly available. While we cannot rule out potential overlap entirely, commonly used graph datasets (Cora, Citeseer, PubMed, Wiki-CS) are likely represented to some extent in the LLM training corpus. However, since our method uses only class labels and general knowledge for OOD generation (rather than direct dataset memorization), we consider the impact of any possible overlap minimal. We further conduct experiments with additional LLMs (GPT-3.5-Turbo and DeepSeek-V3) and observe that pseudo-OOD annotations produced by these LLMs consistently lead to significant improvements in OOD detection performance.

## I    DISCUSSION ON SPECIALIZED DOMAIN APPLICABILITY

In this paper, we primarily evaluate our method on widely used benchmarks. For specialized or less common domains, our approach may require tailored LLM prompting strategies due to differences in domain knowledge. For example, even in well-studied domains, if the ID classes are highly heterogeneous or closely resemble the OOD classes, the ID classifier may fail to learn a clear decision boundary, resulting in high softmax confidence scores for OOD nodes. In this scenario, training the ID classifier solely on ID nodes (as most existing graph OOD detection methods do) leads to poor performance. For instance, on the PubMed dataset, although baselines achieve accurate ID classification, their OOD detection performance is nearly random. Our proposed method leverages pseudo-OOD nodes to provide a more realistic and effective training signal, substantially enhancing the ID classifier's OOD awareness and improving performance by 23.5% over the best-performing baseline. This result suggests that our method offers a greater advantage in OOD detection compared to baselines, particularly in specialized domains such as complex scientific research (e.g., the PubMed dataset). While our method is more effective than prior approaches, we recommend exploring domain-specific adjustments in future work.

## J    THE USE OF LARGE LANGUAGE MODELS (LLMS)

In accordance with the conference policy on LLM usage, we disclose the role of LLMs in this work.

We used OpenAI's ChatGPT both as part of our methodology and as a writing assistant. Methodologically, it was employed for (1) pseudo-OOD node identification, where the LLM classified sampled unlabeled nodes as ID or OOD in a zero-shot setting, and (2) pseudo-OOD node generation, where the LLM produced synthetic OOD node texts. For writing, it assisted with (3) improving clarity, conciseness, and grammar of drafts, (4) suggesting alternative phrasings for technical descriptions, and (5) LaTeX table and figure formatting.

LLMs were not involved in research ideation, experimental design, or result interpretation. All scientific contributions, including problem conception, method development, experimental setup, and analysis, were conducted by the authors. The authors carefully reviewed and verified all LLM outputs (including generated nodes) to ensure accuracy, appropriateness, and originality, and take full responsibility for the final content. LLMs are not considered contributors or authors.

## K    OOD DETECTION METHODS COMPARISON

| Method | Core Technique | Uses LLM? | LLM Role (if any) | Requires Real OOD Data? | Data Type |
|---|---|---|---|---|---|
| GNNSafe Wu et al. (2023) | Post-hoc Energy Scoring | No | N/A | No | Graph |
| GRASP Ma et al. | Post-hoc Score Propagation | No | N/A | No | Graph |
| GNNSafe++ Wu et al. (2023) | OOD Exposure (Real Data) + Energy Regularization | No | N/A | Yes | Graph |
| VOS Du et al. (2022) | Generative Pseudo-OOD (Representations) | No | N/A | No | Image |
| GOLD Wang et al. (2025) | Implicit Adversarial Pseudo-OOD Latent Generation | No | N/A | No | Graph |
| Synthetic Abbas et al. (2025) | LLM-Generated Pseudo-OOD Proxies | Yes | Pseudo-OOD Text Generation | No | Text |
| EOE Cao et al. (2024) | LLM-Envisioned Outlier Exposure | Yes | Outlier Concept Generation | No | Image |
| GOE-identifier | OOD Exposure (LLM-Identified Pseudo-OOD Nodes) | Yes | Pseudo-OOD Node Identification | No | Text-Attributed Graph |
| GOE-generator | OOD Exposure (LLM-Generated Pseudo-OOD Nodes) | Yes | Pseudo-OOD Node Generation | No | Text-Attributed Graph |

Table 11: Comparison of various OOD detection methods.

