# OpenReview forum: "Graph Synthetic Out-of-Distribution Exposure with Large Language Models"
_ICLR.cc/2026/Conference — Submitted to ICLR 2026_

### Official Review · Reviewer_gnhf · 2025-10-27

**Soundness:** 2
**Presentation:** 3
**Contribution:** 2
**Rating:** 2
**Confidence:** 5

**Summary:**

This paper proposes GOE-LLM, a framework for node-level OOD detection on text-attributed graphs (TAGs) that achieves OOD exposure without any real OOD nodes by leveraging LLMs to either identify pseudo-OOD nodes from unlabelled graph nodes (GOE-identifier) or generate synthetic OOD nodes (GOE-generator). Both pipelines can be integrated with an energy-regularised classifier/detector for OOD detection, yielding substantial improvements over baseline methods without exposure and results comparable to approaches that rely on real OOD labels. Across Cora, Citeseer, PubMed, and Wiki-CS, the method achieves up to 23.5% AUROC improvement while maintaining competitive ID accuracy, with extensive ablations on LLM zero-shot annotation quality, number of generated OOD nodes, and LLM variants. There are concerns on the method and experiments.

**Strengths:**

- GOE-LLM provides a clear and simple recipe for LLM-powered synthetic OOD exposure on TAGs that is easy to implement on top of standard GCN training and OOD scoring (e.g., energy score).
- Strong empirical gains over competitive baselines with and without real-OOD exposure, while maintaining ID accuracy.
- The authors have provided thorough practical cost and efficiency analysis, including API call counts, per-dataset costs for GPT-4o-mini, and inference speed.
- Experiments were conducted on multiple LLMs to demonstrate that the method is not tightly coupled to a single proprietary model.

**Weaknesses:**

- The proposed method was evaluated only on four citation-like networks (Cora, Citeseer, PubMed, and Wiki-CS) with limited data scales (500 ID and OOD nodes for testing (Line 272)). These datasets may share similar graph structures (academic citation graphs), textual domains (scientific abstracts), and homophilic connectivity patterns. Such limited coverage may cause domain bias (acknowledged in Appendix but may be insufficiently justified) and hinder the generalisability of GOE-LLM for larger scale evaluations. The evaluation can be greatly improved if the authors included additional well-established TAGs such as Bookhis, Amazon-Photo/Electronics, or Amazon Ratings [1–3], which exhibit different linguistic styles and structural properties.
- Although the authors acknowledge the risk of data leakage between the LLM pretraining corpus and the evaluation datasets, this concern is not rigorously justified or quantified. To evaluate the impact of this, the authors could experiment their approach on temporally newer TAGs such as TAPE-Arxiv-2023 [1] using LLMs whose training cutoff predates that dataset (e.g., GPT-3). This would more convincingly rule out memorisation effects.
- The paper argues that the LLM's noisy annotations (i.e., ID nodes mistakenly labelled as OOD) are beneficial because they "often lie near the ID-OOD boundary and can serve as hard negatives". This is an intriguing hypothesis, but it's presented as a post-hoc justification without any direct evidence (e.g., do they really lie on the boundary?). It is equally plausible that the noise simply acts as a generic regulariser (like label smoothing) that prevents overconfidence, rather than a targeted hard-negative mining strategy.
- The current experiments focus exclusively on label-leave-out OOD settings. However, previous works (e.g., GNNSafe and [3]) have examined structure- and feature-level distribution shifts. It remains unclear whether GOE-LLM is effective or feasible under such conditions. Testing structural perturbations (e.g., edge rewiring, node feature corruption, temporal splits) would better demonstrate general robustness.
- The assumption (Line 226) that the GOE-generator has access to OOD class names is a rather restrictive and strong assumption that weakens the claim of "no real OOD" used and may not hold in open-world scenarios where OOD semantics are unknown.
- The presentation of several figures could be improved. For instance, in Figure 1, it is unclear of the margin when comparing GOE-identifier vs GNNSafe for the PubMed dataset. Intermediate axis-ticks can be added for improved clarity. Additionally, Figure 2 provides an impression that the identifier and generator can be unified together for OOD detection. However, the main text and experimental results treat them as separate, distinct components. This inconsistency should be clarified.

**Questions:**

- While the authors describe multiple strategies for linking pseudo-OOD nodes to the original graph, which specific strategy was employed in the main experiments? Given that embedding-similarity linking may behave differently in homophilic vs heterophilic graphs, did the authors consider or evaluate this factor?
- Is the current training/test split size sufficient to reflect real-world scalability (e.g., using only 500 ID and OOD nodes at test-time), considering modern TAGs often contain tens or hundreds of thousands of nodes (e.g., Arxiv, SportsFit, Products) [1–3]? How would computational and detection performance scale with graph size?
- Can you show some examples or embedding distribution visualisation of the synthetic OOD samples (LLM labelled and LLM generated) vs. Real OOD samples vs. ID samples.
- What was the margins s_id and s_ood used and how are they chosen? Have the authors examined sensitivity to these parameters?
- Have the authors considered distribution shifts beyond purely Label-leave-outs (e.g., structure or feature shifts). If not, could you please comment on how GOE-LLM might perform under such scenarios?
- The authors have noted that "the improvement on the Wiki-CS dataset is much smaller". Do you have any insight as to why? Is there something about that dataset's text or class structure that makes this kind of semantic exposure less effective?
- For the GOE-identifier approach, how was the number of sampled unlabelled nodes (n = 200) selected (Lines 279-280)? What fraction of those nodes were classified as OOD by the LLM, and how does varying this number (e.g., 50 vs 500 nodes) affect detection performance?

[1] He et.al., Harnessing Explanations: LLM-to-LM Interpreter for Enhanced Text-Attributed Graph Representation Learning. ICLR2024.

[2] Chen et.al., Text-space Graph Foundation Models: Comprehensive Benchmarks and New Insights. NIPS2024.

[3] Wang et.al., Text Meets Topology: Rethinking Out-of-distribution Detection in Text-Rich Networks. EMNLP2025.

---

### Official Review · Reviewer_tEUr · 2025-10-28

**Soundness:** 2
**Presentation:** 2
**Contribution:** 2
**Rating:** 2
**Confidence:** 4

**Summary:**

The paper focus on the problem of node OOD detection on TAGs with OOD exposure, to handle the situations where labeled OOD is unavailable, the paper propose to use LLM to 1) identify OOD nodes or 2) generate OOD nodes. Experiments demonstrate that the method achieve competitive OOD detection performance to real OODs.

**Strengths:**

1. Graph OOD detection on TAGs is an important problem while less studied.
2. The paper is clearly formated and easy to understand.
3. Comprehensive experiments are conducted to demonstrate the effectivenss.

**Weaknesses:**

1. As LLM as zero-shot identifier achieves low OOD detection performance (Table 2, 10), it is doubable that it's a good idea to use LLM for OOD exposure.
2. Both the identifer and generator don't consider graph structure, it's unrelated to graph OOD detection.
3. Only class OOD is considered and it can directly be characterized by text semantics, no graph-related OOD patterns are considered.

**Questions:**

1.  In OOD identifier, what's the ID/OOD ratio of 200 randomly sampled nodes, what if the sampled are most ID ones? why choose 200 random samples? What's the influnce of number of sampled nodes?
2. Consider both identifer and generator don't consider graph structure, I wonder why not simply use LMs like BERT as TAG classifier/OOD detector.

---

### Official Review · Reviewer_CMUK · 2025-10-28

**Soundness:** 3
**Presentation:** 2
**Contribution:** 2
**Rating:** 2
**Confidence:** 4

**Summary:**

This paper focuses on node-level graph ood detection. Authors propose to use LLMs to (1) identify potential ood nodes in existing graphs, or (2) directly generate text for pseudo ood nodes. Authors conduct experiment on four text-based graph datasets and verify the effectiveness of the proposed model. However, there are many concerns on the methodology and the experiment. Please refer to below. And the writing is a little bit busy that, there are two relatively unrelated usage of LLMs for potential OOD nodes, and there are two different OOD detectors using either the energy separation or the classification. Authors may need to think about how to effectively organise the paper to have a main focus and deep dive, instead of just combining what has been found effective during the experiment.

**Strengths:**

1.	The utilization of LLMs is interesting, given that the LLMs can deal with the text content in graphs and nodes.

2.	The motivation is clear and convincing in leveraging the generative ability of LLMs to synthesize data.

**Weaknesses:**

1.	In the definition of node-level ood detection, a single-graph semi-supervised transductive scenario is used. This setting is a bit weak for graph learning by avoiding the traditional inductive setting. And there are many scenarios where node-level ood will need to be detected in multi-graph scenario, such as in the Twitch dataset; or a node shift scenario where the unlabelled nodes will possibly replace labelled nodes with a distribution shift in node features as in Text-TopoOOD [1].

2.	The claim in Line 208-212 is too strong: “… we assume a more challenging and realistic scenario where there is no access to any OOD nodes … we leverage the transductive nature of graph learning … identify potential OOD nodes directly from the original graph.” OOD nodes are actually accessible in this setting, although it’s not labelled as OOD nodes.

3.	The design of $A_\text{aug}$ is relatively ad-hoc. Given that the $X^\text{ood}$ are embeddings of synthetic OOD nodes, connecting these synthesized OOD nodes to ID nodes based on embedding similarity sounds very un-straightforward. There should be a more sophistically designed way to link the synthetic OOD nodes to the existing ID nodes.

4.	The arXiv dataset is a very common TAG dataset and OOD detection dataset. Yet it’s not used in this paper. Could the authors give any explanation?

5.	The OOD split is only based on class difference. Yet in existing graph OOD work, many different scenarios are considered, including feature shift, structure shift etc.

6.	The latest baselines are from 2023, the GNNSafe++. Could the authors add any  newer baselines from 2024/2025, like the NodeSafe [2], GOLD [3], TNT-OOD?

7.	For the generated ood node embeddings, authors choose to use SentenceBERT. Is it possible to use LLM embeddings?

8.	Do authors also use SentenceBERT to encode the ID nodes/existing nodes in the datasets? If yes, do other baselines also use this embedding method? If no, how to deal with the dimension match issue? And if no, this is could lead to the fairness and rigor of the comparison.

9.	The reason why the synthetic data model with a (K+1)-class classifier can work well, could be largely due to the OOD split is only based on class split, where all nodes are seen here. If the OOD scenarios become more challenging, like inductive, or other types of shift, the effectiveness of this classifier is doubted. And if it’s structural shift, this classifier could not work properly. Authors could provide experiment in these scenarios.

10.	In Appendix Section E, the prompt template for generating OOD nodes utilize the class information of OOD classes. This is a kind of information leakage. How could an OOD detector know what classes belong to OOD in advance?

11.	In Table 10, the LLM performance is very bad in either ID classification or OOD detection. How could such a poorly-performed model generate useful information for effective OOD detection? Any explanation?

[1] Text Meets Topology: Rethinking Out-of-distribution Detection in Text-Rich Networks, 2025

[2] Bounded and Uniform Energy-based Out-of-distribution Detection for Graphs, 2025

[3] GOLD: Graph Out-of-Distribution Detection via Implicit Adversarial Latent Generation, 2025

**Questions:**

Please refer to the weaknesses.

---

### Official Review · Reviewer_NjbE · 2025-11-01

**Soundness:** 3
**Presentation:** 2
**Contribution:** 2
**Rating:** 4
**Confidence:** 4

**Summary:**

Addressing the out-of-distribution detection challenge in text-attributed graphs , this paper proposes the GOE-LLM framework that eliminates the need for real OOD nodes. Its core lies in leveraging Large Language Models to achieve pseudo-OOD exposure, encompassing two key pipelines: first, identifying pseudo-OOD nodes from unlabeled nodes through zero-shot LLM annotation (referred to as GOE-identifier), and second, generating synthetic pseudo-OOD nodes via LLM prompting and inserting them into the original graph (referred to as GOE-generator). Both approaches optimize the training of the in-distribution classifier using a regularization loss function. Experiments conducted on multiple benchmark datasets such as Cora and Citeseer demonstrate that this framework significantly outperforms existing methods without OOD exposure, achieves performance on par with methods relying on real OOD nodes, and meanwhile maintains excellent ID classification accuracy. It thus provides an efficient and low-cost solution for graph OOD detection in open-world and safety-sensitive applications.

**Strengths:**

1. This paper breaks new ground in the field of graph out-of-distribution (OOD) detection, focusing on a critical unmet need: achieving effective OOD exposure without relying on real OOD nodes. The paper innovatively designs two pipelines: 1. GOE-identifier, which leverages zero-shot LLM annotation to identify pseudo-OOD nodes from unlabeled graph data; 2. GOE-generator, which generates new pseudo-OOD nodes through LLM prompting.

2. Framework is mathematically defined (e.g.，pseudo-OOD exposure loss), node identification/generation formulas) and integrates mature components (2-layer GCN, SentenceBERT, Adam optimizer). Two synthetic OOD model variants (binary detection layer, K+1-class classifier) confirm core method flexibility.Expermental RobustnessExperiments on 4 TAG datasets (Cora, Citeseer, Pubmed, Wiki-CS) with clear ID/OOD splits and standardized train/val/test partitions. Compares with 7 baselines (5 w/o OOD exposure, 2 with real OOD exposure) using 3 metrics (AUROC, AUPR, FPR@95). Ablation studies (pseudo-OOD node count, LLM variants) and reproducibility measures (open code, prompts, 5 random seeds) ensure result reliability.

3. The discussion flows coherently from the presentation of the problem, through the design of the methodology, to the experimental validation. Each chapter is well-structured as distinct modules with clear expression.

4. By replacing real out-of-distribution (OOD) nodes with pseudo-OOD nodes identified or generated by Large Language Models (LLMs), the paper addresses the high cost of acquiring OOD data in practical scenarios. GOE-LLM is cost-effective (e.g., the annotation cost for the Cora dataset is approximately $0.007–$0.02) and efficient (with inference time comparable to post-hoc detection methods), enabling its scalable application in safety-sensitive scenarios.

**Weaknesses:**

1. The framework relies excessively on textual attributes, making it difficult to apply to non-text-attributed graph scenarios. This method can only be used for graphs with rich textual attributes and cannot cover non-text-attributed graph scenarios (such as social networks with only structural/numerical features and biological graphs with molecular structure data). It can be improved by extending GOE-LLM to non-text-attributed graphs.

2. The paper lacks sufficient innovation, as existing studies have already included papers on using large language models to generate outlier nodes in graphs, such as "Out-of-Distribution Detection via LLM-Guided Outlier Generation for Text-attributed Graph".

3. The paper inadequately compares with recent LLM-graph OOD methods and fails to conduct comparisons with concurrent LLM-graph OOD detection or graph GAD (Graph Anomaly Detection) works. This absence makes it difficult for readers to evaluate the innovation and performance advantages of GOE-LLM compared with the latest LLM-graph fusion methods.

**Questions:**

1. You mentioned that the innovation of GOE-LLM lies in leveraging Large Language Models (LLMs) to achieve pseudo-out-of-distribution (OOD) exposure on graphs . However, existing studies—such as the paper "Out-of-Distribution Detection via LLM-Guided Outlier Generation for Text-attributed Graph"—have already explored LLM-based outlier node generation on text-attributed graphs. Could you explicitly clarify: What are the core technical differences between GOE-LLM and this existing work? For example, in terms of pseudo-OOD generation logic (e.g., zero-shot annotation vs. guided generation), integration with graph structure (e.g., how synthetic nodes are embedded into the original graph), or loss function design.

2. The paper acknowledges that the pseudo-out-of-distribution (OOD) nodes identified by Large Language Models (LLMs) are noisy (e.g., the zero-shot OOD annotation accuracy on the Cora dataset is only 0.7190), but it does not propose targeted noise mitigation strategies. Measures such as conducting noise sensitivity analysis (e.g., manually injecting different proportions of label noise into pseudo-OOD annotations) can be implemented.

---

### Meta-Review · Area_Chair_RizN · 2026-01-05

**Summary:**

The paper initially got consistently negative scores: 4, 2, 2, 2. The main weaknesses raised by the reviewers are: limited novelty, lack of comparison with recent methods, the paper is not-well written, without consideration of graph structure, and limited evaluation datasets. The authors did not provide a rebuttal. Thus the AC believes the reviewers will not change their original scores and recommends rejection to this paper.

**Reviewer Concerns:**

The authors did not provide a rebuttal. Thus all the concerns are remained.

**Reviewer Scores:**

The authors did not provide a rebuttal. Thus the reviewers will not change their scores.

---

### Decision · Program_Chairs · 2026-01-26

Reject